# Effect of Transplanting Time and Nitrogen–Potassium Ratio on Yield, Growth, and Quality of Cauliflower Landrace Gigante di Napoli in Southern Italy

Alessio Vincenzo Tallarita [1], Eugenio Cozzolino [2,*], Antonio Salluzzo [3], Agnieszka Sekara [4], Robert Pokluda [5], Otilia Cristina Murariu [6,*], Lorenzo Vecchietti [7], Luisa del Piano [2], Pasquale Lombardi [8], Antonio Cuciniello [2] and Gianluca Caruso [1]

[1] Department of Agricultural Sciences, University of Naples Federico II, Portici, 80055 Naples, Italy; alessiovincenzo.tallarita@unina.it (A.V.T.); gcaruso@unina.it (G.C.)
[2] Council for Agricultural Research and Economics (CREA)—Research Center for Cereal and Industrial Crops, 81100 Caserta, Italy; luisa.delpiano@crea.gov.it (L.d.P.); antonio.cuciniello@crea.gov.it (A.C.)
[3] Territorial and Production Systems Sustainability Department—Research Centre Portici, Italian National Agency for New Technologies, Energy and Sustainable Economic Development (ENEA), 80055 Portici, Italy; antonio.salluzzo@enea.it
[4] Department of Horticulture, Faculty of Biotechnology and Horticulture, University of Agriculture, 31-120 Krakow, Poland; agnieszka.sekara@urk.edu.pl
[5] Department of Vegetable Growing and Floriculture, Faculty of Horticulture, Mendel University, 613 00 Brno, Czech Republic; robert.pokluda@mendelu.cz
[6] Department of Food Technology, University 'Ion Ionescu de la Brad' of Life Sciences of Iasi, 700490 Iasi, Romania
[7] Hydro Fert S.r.l., 76121 Barletta, Italy; l.vecchietti@hydrofert.it
[8] Research Center for Vegetable and Ornamental Crops, 84098 Pontecagnano Faiano, Italy; pasquale.lombardi@crea.gov.it
* Correspondence: eugenio.cozzolino@crea.gov.it (E.C.); otilia.murariu@iuls.ro (O.C.M.)

**Abstract:** Research has been increasingly focusing on the preservation of the biodiversity of vegetable crops under sustainable farming management. An experiment was carried out in southern Italy on *Brassica oleracea* L. var. botrytis, landrace Gigante di Napoli, to assess the effects of two transplanting times (9 September and 7 October), in factorial combination with five nitrogen–potassium ratios (0.6; 0.8; 1.0; 1.2; and 1.4) on plant growth, yield, and quality of cauliflower heads. A split-plot design was used for the treatment distribution in the field, with three replications. The earlier transplant and the 1.2 N:K ratio led to the highest yield, mean weight, and firmness of cauliflower heads which were not significantly affected by both transplanting time and N:K ratio in terms of colour components. The 1.2 N:K ratio led to the highest head diameter with the earlier transplant, whereas the 1.0 ratio was the most effective on this parameter in the later crop cycle. The highest nitrate, nitrogen, and potassium concentrations in the heads were recorded with the earlier transplanting time. Antioxidant activity, ascorbic acid, and polyphenol content increased with the rise of the N:K ratio. The element use efficiency was constantly negative with the N:K increase for nitrogen and was augmented until the 1.2 ratio for potassium. The results of our investigation showed that the optimal combination between transplanting time and N:K ratio is a key aspect to improve head yield and quality of the cauliflower landrace Gigante di Napoli, under the perspective of biodiversity safeguarding and valorisation.

**Keywords:** local variety; *Brassica oleracea* var. botrytis; antioxidant activity; ascorbic acid; polyphenols

## 1. Introduction

The preservation of agricultural biodiversity encompasses a multitude of practices aimed at sustaining the genetic variability of plants, especially those utilised in food production [1]. Within the latter context, the cauliflower landrace (*Brassica oleracea* L. var. botrytis) Gigante di Napoli is a traditional crop in the Campania region (Italy), with an

important cultural heritage and cultivation practices traditionally rooted in local agriculture. It is appreciated for its organoleptic quality, high texture, colour ranging from white to cream of the heads, delicate taste, and versatility in cooking [2,3].

From the agronomic point of view, Gigante di Napoli requires specific choices to express its full development potential, including the most appropriate transplanting time and adequate fertilisation management, particularly referring to the nitrogen/potassium ratio. The latter aspects influence both the yield and quality of the final product and the preservation of the global characteristics of this landrace [4,5].

The cultivation and promotion of Gigante di Napoli are essential either to keep alive a part of Italian agricultural and cultural heritage or to offer farmers an opportunity to diversify and sustainably manage the crop system. In order to make it easier and feasible for farmers to choose and cultivate local varieties, it is crucial to help them identify, under the current climate change conditions, the best fitting transplanting time and fertilisation methods.

Transplanting time is a key factor that significantly influences plant growth, and crop yield and quality [6,7]. The choice of the optimal transplanting time is affected by several factors, including genotype, climatic and cultivation requirements [8,9]. Based on cauliflower characteristics for germination, growth, and ripening [10], temperatures under 10 °C can slow down its development, while temperatures higher than 25 °C may induce the premature 'sprout' of the heads, compromising the quality of the final product [11]. Moreover, the choice of the appropriate transplanting time allows for a more efficient use of water and nutrient resources [12], both by reducing the need for supplementary irrigation and identifying the phases of the most efficient plant nutrient uptake, thus optimising the fertilisation.

The ratio between nitrogen (N) and potassium (K) in the soil plays a crucial role in plant nutrition and directly influences crop yield and quality [13]. The two mentioned elements are among the primary nutrients needed by plants and have complementary but distinct functions that, when properly balanced, can lead to optimal results in terms of yield and quality. In fact, nitrogen is essential for the synthesis of proteins, nucleic acids, and chlorophyll, promoting leaf development and plant tissue formation through its key function in photosynthesis. Adequate nitrogen availability in the soil supports rapid and vigorous growth, but an excess of this mineral element may lead to excessive leaf growth which slows and reduces flowering and/or fruiting, as well as potential negative impacts on the environment, such as the eutrophication of surface waters [14,15].

Potassium is vital for osmotic regulation, enzyme activation and sugar transport, contributing to plant resistance to stress, improving their ability to withstand adverse conditions such as drought, cold, and disease [16]. This element is also crucial to fruit quality, influencing attributes such as colour, flavour, firmness, and storability [17]. The most appropriate N:K ratio is therefore essential for balancing vegetative growth with the reproduction phase and stress resistance, with consequent positive effects on yield and crop quality. Determining the optimal ratio depends on the genotype, and the specific environmental conditions of the cultivation area [18,19].

This study aimed to investigate the effects of transplanting time and nitrogen/potassium ratio on plant growth, yield, and quality of cauliflower landrace Gigante di Napoli in the Campania region (southern Italy), to highlight the importance of tailored agricultural practices in maintaining crop diversity, thus supporting sustainable agriculture and food security in the face of climate change.

## 2. Materials and Methods

### 2.1. Plant Material, Growing Conditions, and Experimental Protocol

Research was conducted at the Department of Agricultural Sciences, University of Naples Federico II, Naples, Italy (40°500 N, 14°150 E, 17 m a.s.l., in a Mediterranean or Csa climate according to the Köppen classification scheme) [20] on cauliflower (*Brassica oleracea* (L.) var. botrytis, landrace Gigante di Napoli), in open fields, in 2021 and 2022. Cauliflower

plants were grown in a sandy loam soil and the roots developed in the superficial 40 cm profile having 77% sand, 16% silt, and 7% clay, and 2.3% organic matter content ($w/w$), 0.13% total nitrogen, 33.7 mg kg$^{-1}$ P, and 1369 mg kg$^{-1}$ exchangeable K. The average monthly total rainfall and temperature were the following: 10 mm and 25 °C in September; 20 mm and 22 °C in October; 90 mm and 18 °C in November; 70 mm and 14 °C in December; and 60 mm and 10 °C in January. The plants were spaced 50 cm along the rows which were 70 cm apart, with a density of 2.9 plants per m$^2$.

The experimental protocol was based on the factorial combination of two transplanting times (9 September, 7 October) and five nitrogen/potassium ratios (0.6; 0.8; 1.0; 1.2; and 1.4), using a split-plot design for the treatment distribution in the field, with three repetitions, with the experimental unit covering a 10.5 m$^2$ (3.5 × 3.0 m) surface area. The N:K ratios were applied by keeping constant the potassium dose and changing the nitrogen one, based on the following average cauliflower requirements: 250 kg ha$^{-1}$ N, 60–100 kg ha$^{-1}$ P$_2$O$_5$, and 250 kg ha$^{-1}$ K$_2$O. The mentioned fertilisations were achieved by supplying calcium nitrate (15.5 N, 23.0 CaO), potassium nitrate (13 N, 46 K$_2$O), potassium sulphate (50 K$_2$O, 45 SO$_3$), and phosphoric acid (54% P$_2$O$_5$), partly during pre-transplanting (35%) and the remainder during the crop cycles (65%).

Drip irrigation was activated when the soil available water capacity (AWC) decreased to 70%. The harvests were carried out when the cauliflower heads completed their growth, cutting the whole plants of the central rows at 2 cm above soil level, excluding the border ones, on 21 November and 27 December, for the earlier and the later transplanting times, respectively, as an average of 2021 and 2022.

At harvest, 15-plant samples were randomly collected in each plot to assess the fresh and dry weight (in an oven at 70 °C under vacuum, at 15 kPa pressure, and until they reached constant weight) of the cauliflower heads.

For each sample, firmness (digital penetrometer with 8 mm tip T.R. Turoni s.r.l., Forli, Italy) and colour (L\*, a\*, b\*—CIElab), using a Minolta CR-300 Chroma Meter (Minolta Camera Co. Ltd., Osaka, Japan), were measured.

### 2.2. Nitrate and N, P, and K Determinations

Head samples of 0.5 g per treatment were ground using an IKA mill (IKA-Werke, Staufen, Germany), sieved through a 2 mm screen, to measure nitrate content by a Foss continuous flow Analyzer—FIAstar 5000 (FOSS Italia S.r.l., Padova, Italy), as reported by Di Mola et al. [21]. An aliquot of the samples was washed with ultrapure water, dried, and stored in the freezer (−80 °C) to perform the following analyses: ascorbic acid, total phenols, and antioxidant activity. Frozen 'curds' were roughly cut with a ceramic blade knife, brought back to −20 °C, and homogenized in a knife mill Grindomix GM300 Retsch (Haan, Germany). The grinding program was in two steps: 1000 rpm revolution speed, direction impact, for 20 s (pre-grinding) and 2000 rpm revolution speed, direction cut, for 30 s (fine grinding). The homogenate obtained was divided into sub-samples and stored at −80 °C and used for the determination of ascorbic acid, total phenols, and antioxidant activity.

### 2.3. Ascorbic Acid

Ascorbic acid extraction procedure (in duplicate for each sample) was conducted according to the AOAC's *Official Methods of Analysis*. Briefly, 10 mL of extraction solution (30 g L$^{-1}$ MPA—80 mL L$^{-1}$ acetic acid—1 mmol L$^{-1}$ EDTA) were added to 3–4 g of homogenate and the mixture was shaken for 3 min with a disperser (Ultra-TuraxT25 IKA-Werke GmbH & Co. KG, Staufen, Germany). The resulting extract after centrifugation (10,000 rpm; 20 min; 4 °C) was immediately analysed by the colorimetric method using the Folin Phenol Reagent according to Jagota and Dani [22]. To 400 μL of the sample, 200 μL of extraction solution, 1.2 mL of ultrapure water, and 200 μL of Folin–Ciocalteu's phenol reagent 2N (diluted 1:5 with ultrapure water) were added. After 30 min in darkness and at room temperature, the absorbance of the blue colour developed was measured at 760 nm.

This procedure was repeated in triplicate. For quantification, six standard solutions of ascorbic acid (range 25–150 µg mL$^{-1}$) were prepared and estimated as above (R$^2 \geq$ 0.996).

### 2.4. Total Phenols

Total phenol extraction was performed following the procedure as described by Kaur et al. [23] with minor modifications. Approximately 5 g of sample (in triplicate) were homogenized in 25 mL of 80:20 CH$_3$OH/H$_2$O mixture and sonicated in an ultrasonic bath (Elmasonic P Elma, Singen, Germany) at 40 °C for 3 h in darkness. Afterwards, the extracts were centrifuged for 20 min at 10,000 rpm (HF 14.94 rotor, radius 9.7 1086 RCF) at 4 °C (Centifuge 17R Heraeus Sepatech, Waltham, MA, USA) and the upper layer was separated and kept at −20 °C until analyses were performed. The total phenolic content was determined according to Folin–Ciocalteu's method [24] with small modifications. In the test tube, the following were added in sequence: 1 mL H$_2$O, 100 µL of cauliflower extract, 100 µL of Folin–Ciocalteu's phenol reagent 2 N solution, and, after 10 min, 800 µL of Na$_2$CO$_3$ 75 g L$^{-1}$ solution, so that the final pH was $\geq$10 (test final volume 2 mL). The mixture was allowed to stand for 120 min at room temperature and in darkness, and absorption was measured at 765 nm against a reagent blank. Measurements were performed in duplicate, and results were expressed as mg of gallic acid equivalent per 100 mg of fresh weight (GAE 100 g$^{-1}$ f.w.). For calibration, six standard solutions of gallic acid (range 0–150 µg mL$^{-1}$) were prepared and estimated as above (R$^2 \geq$ 0.998).

### 2.5. Antioxidant Activity

The antioxidant activity was measured on the cauliflower extract prepared as for total phenolic determination, using the DPPH assay based on the measurement of the scavenging ability of the antioxidant toward the stable radical DPPH. A 3.9 mL aliquot of a 0.0634 mM of DPPH solution in methanol was added to 0.1 mL of each extract and shaken vigorously. The change in the absorbance of the mixture at 515 nm was observed and the antiradical activity of the sample was evaluated from the percentage of DPPH remaining when the kinetics reached a steady state. It has been found experimentally that for cauliflower fruit, this occurs after 150 min. A stock solution of DPPH 634 µM (12.5 mg in 50 mL of degassed methanol) was prepared at the assay moment. For the test, instead, a 10-fold fresh diluted solution is used. Dark glass test tubes were used with Teflon-coated rubber septa and aluminium caps (Supelco Inc., Bellefonte, PA, USA), and the DPPH solution was degassed and flushed with helium gas for 3 min. Trolox equivalency is used as a benchmark for the oxidant capacity of the extracts. Therefore, the calibration curve was constructed using nine standard solutions containing this antioxidant (range 0.15–20 µM) and the results are expressed as µmol of Trolox equivalent per 100 g of fresh weight (µmol TE 100 g$^{-1}$ f.w.).

### 2.6. Nitrogen Use Efficiency (NUE) and Potassium Use Efficiency (KUE) on Yield Calculation

For nitrogen use efficiency (NUE) and potassium use efficiency (KUE) on yield, the following formulas, based on average values referred to the variable examined per level of N:K treatment, were adopted:

$$\text{NUE} = \frac{\text{Dry weight of crop} \left(\text{kg ha}^{-1}\right)}{\text{Amount of nitrogen applied} \left(\text{kg ha}^{-1}\right)} \qquad \text{KUE} = \frac{\text{Dry weight of crop} \left(\text{kg ha}^{-1}\right)}{\text{Amount of potassium applied} \left(\text{kg ha}^{-1}\right)}$$

### 2.7. Statistical Analysis

Data were processed by analysis of variance (two-way ANOVA) and mean separations were performed through the Tukey's test, with reference to 0.05 probability level, using the SPSS software version 29.

## 3. Results and Discussion

### 3.1. Yield and Growth Parameters

No significant differences arose between the two research years (2021 and 2022) regarding the variables examined and, therefore, the average values were reported.

The later transplanted crops exhibited a longer growing period, compared to the earlier ones (81.4 vs. 73.1 days), resulting in significantly higher yields (11.2 vs. 9.0 t ha$^{-1}$), as observed in Table 1.

**Table 1.** Effect of transplanting time and N:K ratio on yield and plant growth parameters of cauliflower.

| Experimental Treatment | Crop Cycle Duration (Days) | Yield (t ha$^{-1}$) | Head Fresh Weight (kg) | Head Dry Weight (kg) | Firmness (N) | Colour | | |
|---|---|---|---|---|---|---|---|---|
| | | | | | | L* | a* | b* |
| Transplanting time | | | | | | | | |
| 9 September | 73.1 b | 11.1 a | 0.83 a | 0.17 | 6.65 a | 85.9 | 0.05 | 0.65 |
| 7 October | 81.4 a | 9.0 b | 0.61 b | 0.16 | 5.44 b | 85.8 | 0.54 | 0.04 |
| | | | | n.s. | | n.s. | n.s. | n.s. |
| N:K ratio | | | | | | | | |
| 0.6 | 74.8 b | 9.2 d | 0.41 d | 0.14 c | 4.71 d | 85.5 | 0.44 | −0.10 |
| 0.8 | 75.1 b | 9.1 d | 0.47 c | 0.15 c | 5.81 c | 86.1 | −1.04 | 0.85 |
| 1.0 | 76.9 ab | 11.0 b | 0.76 b | 0.17 b | 6.45 b | 87.0 | 0.62 | 0.83 |
| 1.2 | 79.7 a | 11.3 a | 1.10 a | 0.19 a | 6.96 a | 85.6 | 0.81 | −0.17 |
| 1.4 | 79.8 a | 9.8 c | 0.86 b | 0.17 b | 6.31 b | 84.8 | 0.40 | 0.31 |
| | | | | | | n.s. | n.s. | n.s. |

n.s. not significant; within each column, mean values followed by different letters are significantly different at $p \le 0.05$ according to Tukey's test.

No significant interactions between the two experimental factors on the variables examined arose, with the exception of the diameter of cauliflower heads (Figure 1), which was the highest corresponding to 1.2 N:K with the earlier transplant and to 1.0 with the later one. The earlier transplanting time also led to a higher mean head fresh weight (0.83 kg), compared to the later one (0.61 kg). However, no significant differences were detected in the dry weight, indicating that the increased biomass in the earlier crops was due to the higher water content. Furthermore, the head firmness associated with the 9 September transplant was higher (6.65 N) as compared to the later crop cycle (5.44 N). The colour parameters (L*, a*, b*) were not significantly affected by the transplanting time.

The 1.2 and 1.4 N:K ratios led to a longer crop cycle, compared to 0.6 and 0.8. The highest yield was recorded at 1.2 N:K (11.3 t ha$^{-1}$), due to the highest head fresh weight (1.10 kg), and the lowest at 0.6 ratio (0.41 kg).

Both the head dry weight and firmness reached the highest value at 1.2 N:K (0.19 kg and 6.96 N, respectively) and the lowest at 0.6 (0.14 kg and 4.71 N, respectively).

The colour components (L*, a*, b*) were not significantly affected by the N:K ratio.

In a previous investigation [25], the transplanting time significantly influenced the growth and yield of cauliflower, with the transplant practiced at the end of November enhancing biometrical parameters and yield, particularly plant height, leaf number, stem diameter, and head size.

Furthermore, Kartika et al. [26] reported the detrimental effects of delayed transplanting time on yield and plant growth indicators, such as leaf number, canopy area, and curd characteristics. Moreover, Sultana et al. [27] found that the optimal transplanting time to maximize growth and yield also depends on cultivar and latitude.

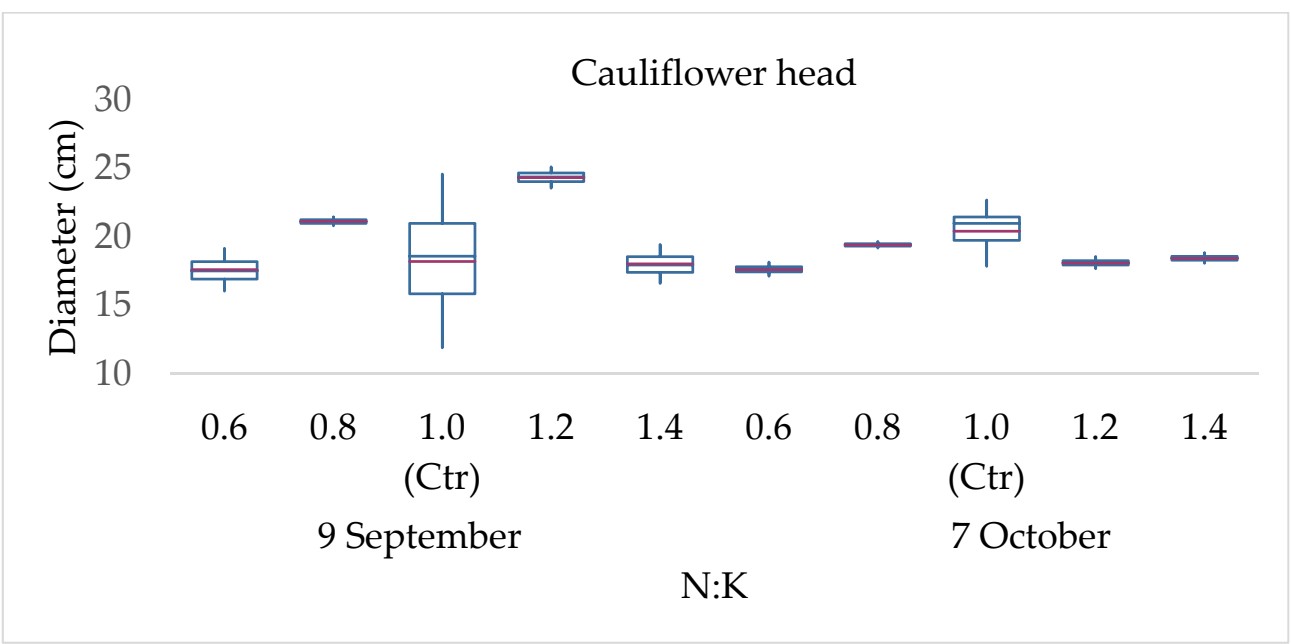

**Figure 1.** Interaction between transplanting time and N:K ratio on cauliflower head diameter.

Cauliflower plants experience physiological and biochemical reactions when subjected to cold temperatures, encompassing acclimatisation which allows them to progressively adapt to the modified environment, thus enhancing their ability to benefit from the mentioned stress [28] resulting in a remarkable rise in marketable yields. In this respect, the adaptation to low temperatures involves several phenomena, including alterations in protein synthesis, variations in membrane composition, and activation of antioxidant defence systems.

Rahman et al. [29] highlighted the relationship between planting time and plant hormonal status, which affects cauliflower production. Wadan et al. [30] investigated the effects of nitrogen/potassium ratios on yield, revealing that a combination of 120 kg N and 90 kg K per hectare significantly enhanced yield and growth parameters, including head weight and leaf area.

### 3.2. Nitrate, Nitrogen, Phosphorus, and Potassium Concentrations

In a study conducted by Yildirim [31], it was shown that raising the amount of nitrogen fertiliser in the rhizosphere led to an increase in the availability of other minerals in the soil solution which, in turn, resulted in improved growth and higher yield and weight of cauliflower heads. Further, the amount of nitrogen supply is reportedly crucial for plant development as well as nitrate absorption and accumulation [32].

Marschner [33] emphasized the significance of the nitrogen–potassium relationship in plant metabolism and the potential consequences of imbalances, i.e., a decreased rate of photosynthesis and nitrogen absorption, as well as the influence of the ratio between the two elements in the soil on their absorption.

Nitrogen has a crucial role in promoting plant growth as it is a vital constituent of chlorophyll. Probably, the higher nitrogen availability resulted in higher chlorophyll levels, which improves the plant capacity to convert light into energy for its growth and development. This hypothesis is consistent with the reports of Katuwal et al. [34] who found that cauliflower plants subjected to different nitrogen levels showed an increase in head fresh weight, leaf area, and plant height, upon the augmentation of nitrogen supply. The latter is notoriously essential for the synthesis of proteins, which have a crucial role in the organization and functioning of plant cells, including the enzyme enhancement of all metabolic processes.

The influence of transplant nutrient conditioning (TNC) and the nitrogen–potassium (N:K) ratio on the growth characteristics, and on cauliflower diameter, was also reported in previous study [31], elucidating that TNC at different levels of N, P, and K significantly impacted early growth traits such as stem diameter in cauliflower seedlings.

An increase in TNC levels of N enhanced growth but also induced greater transplant shock, which was compensated by faster growth in later stages, suggesting a critical balance between nutrient provision and the plant stress response.

Further confirming the pivotal role of nitrogen, Katuwal et al. [34] demonstrated that the application of 130 kg ha$^{-1}$ N improved yield and biometrical parameters, including stem diameter and head girth.

Similarly, Cutcliffe and Munro [35] reported that cauliflower yield was significantly augmented by nitrogen and phosphorus applications, optimally at N applications ranging from 112 to 224 kg ha$^{-1}$.

Wadan et al. [30] further reported that fertilizer levels of 120 kg ha$^{-1}$ N plus 90 K kg ha$^{-1}$ substantially enhanced yield performance in terms of head weight and diameter and leaf area. The latter finding indicates that the appropriate N:K ratio is crucial to achieve optimal growth and head size in cauliflower production.

As shown in Table 2, the earlier transplanting time led to higher concentrations of nitrate and nitrogen content in cauliflower heads (341 and 3728 ppm, respectively), compared to the later transplant (278.9 and 3050 ppm, respectively). Potassium accumulated at a significantly higher extent in the heads for the earlier transplanting time compared to the later one.

**Table 2.** Effect of transplanting time and N:K ratio on quality parameters of cauliflower heads.

| Experimental Treatment | Nitrate (mg kg$^{-1}$) | N (mg kg$^{-1}$) | P (mg kg$^{-1}$) | K (mg kg$^{-1}$) |
|---|---|---|---|---|
| Transplanting time | | | | |
| 9 September | 341.00 a | 3728.27 a | 1081.25 | 4454.17 a |
| 7 October | 278.90 b | 3050.41 b | 884.62 | 3644.32 b |
| | | | n.s. | |
| N:K ratio | | | | |
| 0.6 | 393.90 a | 3706.25 a | 1016.21 a | 5816.14 a |
| 0.8 | 342.23 b | 3483.88 b | 1036.50 a | 4942.54 b |
| 1.0 | 303.21 c | 3449.48 b | 999.52 a | 3105.98 c |
| 1.2 | 273.28 d | 3157.52 c | 958.75 ab | 3378.85 c |
| 1.4 | 237.36 e | 3149.57 c | 903.59 b | 3002.73 d |

n.s. not significant; within each column, mean values followed by different letters are significantly different at $p \leq 0.05$, according to Tukey's test.

Increasing the nitrogen–potassium ratio, a decreasing trend of nitrate and nitrogen concentrations in cauliflower heads was recorded. The lowest nitrogen–potassium ratio (0.6) resulted in the highest nitrate content (393.9 ppm), which gradually decreased up to the 1.4 ratio (237.4 ppm), and a similar trend was shown by nitrogen. Phosphorus content was the highest at the lowest nitrogen–potassium ratio, whereas potassium content decreased with increasing N:K ratio (from 5816.1 ppm at 0.6 to 3002.73 ppm at 1.4).

Nitrate reduction is a stage in nitrate assimilation affecting the latter process [36]; nitrate reductase is the related enzyme that is activated in response to light conditions, and a strong correlation exists between the activity of nitrate reductase and the concentration of nitrate in plants [37]. In addition, the expression of nitrate absorption and reduction systems is induced by both endogenous nitrogen and plant development [38]. According to Chen et al. [39], the growth of cabbage plants is strongly influenced by the amount of nitrogen they receive. When the nitrogen supply exceeded the optimal threshold, the plants experienced a decrease in growth and an increase in nitrate accumulation leading to toxicity symptoms caused by high levels of $NO_3^-$ concentration, which in turn caused a

biomass decrease. However, over the mentioned threshold, there was a slower rate of rise in nitrate concentration, maybe due to a reduction in nitrate absorption as a consequence of a decrease in plant growth, which also resulted in yield loss. The efficiency of the net nitrate absorption rate significantly improves with the increase of plant relative growth rate, as observed by Ter Steege et al. [40], which highlights the critical role of plant growth in nitrate uptake. While most studies have demonstrated that nitrate reductase is an enzyme induced by its substrate [41], a few investigations have shown that even a small amount of nitrate is enough to trigger nitrate reductase induction [42]. Additionally, the expression of nitrate reductase may be regulated by either the flow of nitrogen or the nitrogen status of plants [43]. In the trials conducted by Chen et al. [39], it was shown that nitrate had a beneficial impact on nitrate reductase activity only when the nitrate supplies were low. However, when the nitrogen inputs were at their optimal level, the nitrate reductase activity either achieved a plateau value or even dropped.

Recent research elucidated the intricate dynamics between nitrogen, phosphorus, and potassium applications and their effects on cauliflower growth, yield, maturity, and nutrient content. In fact, Sony et al. [44] highlighted the significant N-P interaction on growth parameters and yield, suggesting that tailored N-P management can strongly impact nitrate, N, P, and K accumulation in plants whereas, in contrast, in Everaarts' [45] investigation, the increased N application rates did not augment the nitrogen content.

Metwaly [46] studied the nutrient dynamics by examining the effects of P and K on cauliflower, demonstrating that the relationship between these nutrients significantly affected vegetative growth, leaf chemistry, and head yield and quality, thereby emphasizing the importance of optimizing P and K levels along with N management to enhance cauliflower production and nutrient profile optimization.

*3.3. Quality Parameters*

The later transplant elicited a higher concentration of ascorbic acid (4.6 mg 100 g$^{-1}$ f.w.) and total phenols (78.4 mg equivalent GAE 100 g$^{-1}$ f.w.) compared to the earlier one (3.7 mg 100 g$^{-1}$ f.w. for ascorbic acid and 64.2 mg equivalent GAE 100 g$^{-1}$ f.w. for total phenols; Table 3). The antioxidant activity did not show a significant difference between the two transplanting times.

**Table 3.** Effect of transplanting time and N:K ratio on ascorbic acid, antioxidant activity, and total polyphenols of cauliflower curds.

| Experimental Treatment | Ascorbic Acid (mg 100 g$^{-1}$ f.w.) | Antioxidant Activity (μmol Trolox Equivalent 100 g$^{-1}$ f.w.) | Total Phenols (mg Equivalent GAE 100 g$^{-1}$ f.w.) |
|---|---|---|---|
| Transplanting | | | |
| 9 September | 3.74 b | 368.06 | 64.17 b |
| 7 October | 4.55 a | 362.97 | 78.43 a |
| | | n.s. | |
| N:K | | | |
| 0.6 | 3.37 c | 330.44 b | 55.57 c |
| 0.8 | 3.95 b | 368.69 ab | 67.25 b |
| 1.0 | 3.92 b | 346.49 b | 69.28 b |
| 1.2 | 4.88 a | 387.42 ab | 79.78 ab |
| 1.4 | 4.63 a | 394.50 a | 84.66 a |

n.s. not significant; within each column, mean values followed by different letters are significantly different at $p \leq 0.05$ according to Tukey's test.

The ascorbic acid content was the lowest at the 0.6 ratio (3.4 mg 100 g$^{-1}$ f.w.) and increased progressively with a rising nitrogen–potassium ratio, peaking at 1.2 with 4.9 mg 100 g$^{-1}$ f.w. The antioxidant activity and total polyphenols also showed an increasing trend from the lowest to the highest nitrogen–potassium ratio. Both the highest antioxidant activity and total polyphenol content were recorded at 1.4 (394.5 μmol Trolox equivalent 100 g$^{-1}$ f.w. and 84.7 mg equivalent GAE 100 g$^{-1}$ f.w.).

The mentioned findings reflect the favourable conditions associated with delayed transplanting and an increased nitrogen–potassium ratio for enhancing the phytochemical content of plants, augmenting their nutritional value and health benefits.

Some studies highlighted the significant impact of transplanting time and the applied nitrogen–potassium (N:K) ratio on the vitamin C content, antioxidant activity, and polyphenol levels in cauliflower [47]. Contrary to the trend recorded in our study, Lisiewska and Kmiecik [47] found that the increasing nitrogen fertilization decreased the vitamin C content in cauliflower. Furthermore, Giri [48] reported that a lower dose of nitrogen, when combined with organic manures, improved the post-harvest quality of cauliflower, including higher vitamin C content, thus valorising sustainable fertilization strategies. Šlosár et al. [49] observed that nitrogen and sulphur fertilization is a key factor to enhance the accumulation of health-promoting compounds, positively affecting the content of vitamin C and β-carotene in cauliflower. Moreover, a previous study carried out on coloured potato cultivars [50], reported that nitrogen was more effective than potassium in increasing anthocyanin content, and the modulation of the ratio between these two elements was an effective way to increase the expression of polyphenolic compounds.

### 3.4. Nitrogen and Potassium Use Efficiency

In Figure 2, the trend of nitrogen and potassium use efficiency related to the average yield of cauliflower heads is illustrated. The nitrogen use efficiency showed a negative trend from the lowest to the highest nitrogen dose applied. Differently, the potassium use efficiency displayed a slightly positive trend from 0.8 to 1.2 (N:K), then decreasing from 1.2 to 1.4. The decreasing trend of nitrogen use efficiency (NUE) with increasing N:K ratios, along with a stable potassium use efficiency (KUE), might be due to a complex interplay of factors. Plants require a balanced nutrition which can be altered by an excess of nitrogen supplied, especially if potassium, which is crucial for stress tolerance and enzyme activation, is not increased proportionally. The mentioned imbalance could lead to excessive consumption of nitrogen, causing plant absorption to be higher than needed which does not turn out into an increased head yield, thus leading to inefficiency. The relatively stable efficiency of potassium suggests that this element was remarkably used by the plants, indicating its sufficiency across the ratios. Furthermore, nutrient competition at the root uptake sites may limit the availability and absorption of some elements, which are also influenced by the soil properties, such as pH and microbial activity.

Nitrogen use efficiency (NUE) is an essential parameter to measure the efficiency of plant utilization of available nitrogen (N) for growth and yield, encompassing the processes of N uptake, assimilation, and use within a plant system. Studies of different plant species and environmental conditions reveal that NUE is a complex trait influenced by both genetic and environmental factors, showing variability under different levels of nitrogen supply [51]. In *Helianthus annuus*, a trade-off between nitrogen productivity (NP) and mean retention time of nitrogen (MRT) suggests that plants optimize their nitrogen use strategies based on available N levels, either by maximizing nitrogen conservation or utilization [51]. At the molecular level, as observed in *Brassica napus* under low nitrogen (LN) conditions, NUE is enhanced through the regulation of root architecture and nitrogen transporter activity, specifically through the upregulation of nitrate transporters like BnNRT1.5 and downregulation of BnNRT1.8, facilitating efficient nitrate uptake and translocation. These adjustments are part of a plant's adaptive response to varying N availability, aiming to maintain growth and yield with minimal nitrogen input, reflecting an intricate balance between the demand for nitrogen and its availability in the environment [52].

According to previous works [46,53], the combined effects of nitrogen and potassium enhance the growth and yield of cauliflower by increasing the uptake of nutrients, such as nitrogen for protein synthesis, and potassium for regulating water use through stomatal conductance, and enhancing resistance to stress. Once the optimal point is reached, any additional increases in N and K concentrations do not result in further growth or yield advantages, maybe because of the limitation of the plant's capacity to uptake nutrients, in-

efficiencies in water usage and stomatal regulation, and imbalances in nutrient distribution. The mentioned factors reflect a delicate balance between nutrient, water, and physiological requirements for maximizing plant yield.

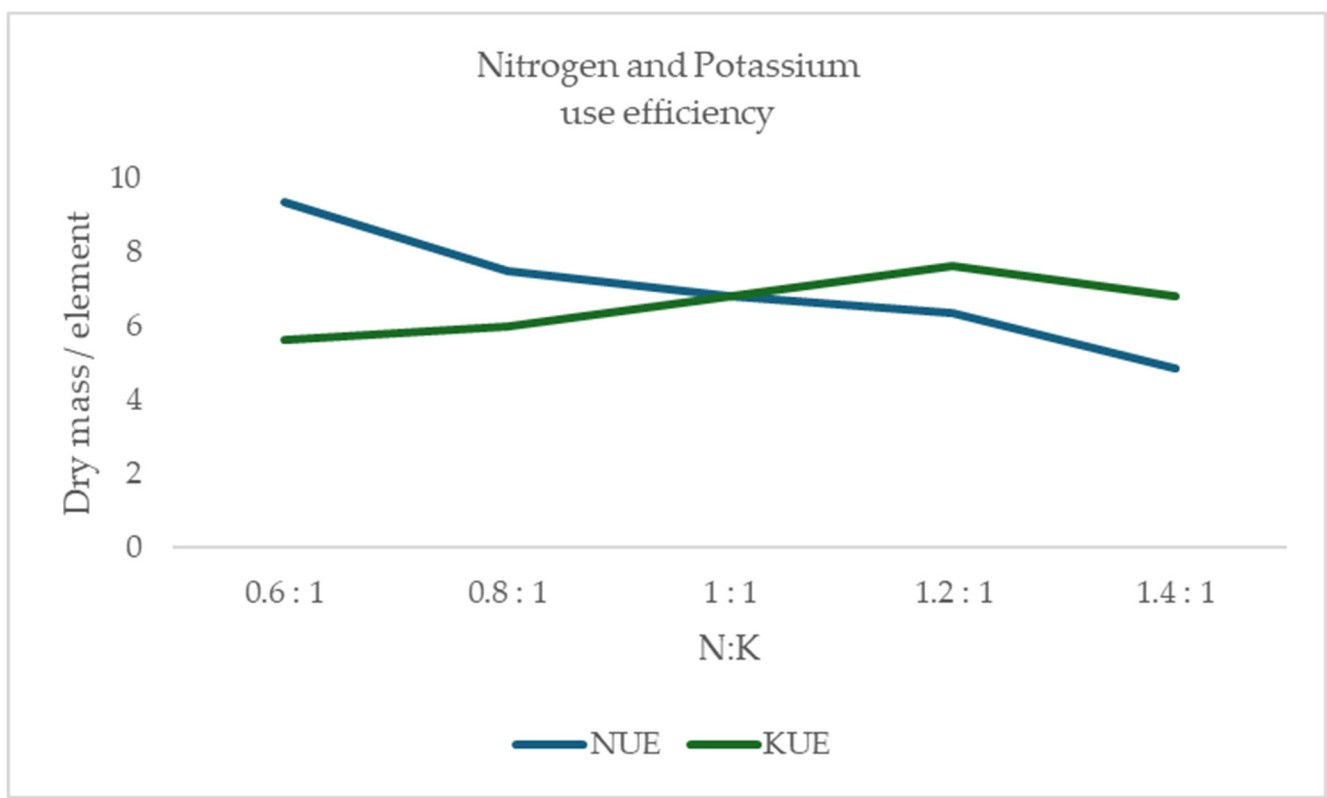

**Figure 2.** Nitrogen use efficiency (NUE) and potassium use efficiency (KUE) in relation to N:K.

### 4. Conclusions

From this research it can be inferred that the choice of both the most appropriate transplanting time and nitrogen–potassium ratio is crucial for the cauliflower landrace Gigante di Napoli, under the perspective to sustain 'niche' crops that are useful both for the economy closely linked to an agricultural area and for its biodiversity. The earlier transplanting time (9 September) proved to be the most effective for head yield, while the later one (7 October) for the quality characteristics. A nitrogen–potassium ratio above 1.2 was not effective to enhance either yield or curd quality. Nitrogen and potassium use efficiency showed different trends, with the first one decreasing with the augmentation of nitrogen supplied, and the second one increasing from 0.8 to 1.2 N:K ratio. From this study, a significant effect of the nitrogen–potassium ratio arose on the yield and quality of the cauliflower landrace Gigante di Napoli, whose importance closely relates to biodiversity and land preservation. The latter aim should be pursued, targeting the compromise between efficient plant nutrition and territory safeguarding, and choosing the best fitting transplanting time both to enhance production and face climate changes.

**Author Contributions:** Conceptualization, A.V.T. and G.C.; methodology, A.V.T., A.S. (Antonio Salluzzo), A.S. (Agnieszka Sekara), O.C.M. and L.d.P.; software, A.V.T., E.C. and P.L.; validation, R.P. and G.C.; formal analysis, A.V.T., E.C., A.S. (Antonio Salluzzo) and L.d.P.; investigation, A.V.T., E.C., L.d.P., P.L. and A.C.; data curation, A.V.T., E.C. and A.S. (Antonio Salluzzo); writing—original draft preparation, A.V.T.; writing—review and editing, A.S. (Agnieszka Sekara), R.P., O.C.M. and G.C.; visualization, A.V.T.; supervision, G.C.; project administration, A.V.T., L.V. and G.C.; funding acquisition, L.V. and G.C. All authors have read and agreed to the published version of the manuscript.

**Funding:** This research received no external funding.

**Data Availability Statement:** Data is contained within the article.

**Conflicts of Interest:** Author Lorenzo Vecchietti was employed by the company Hydro Fert S.r.l. The remaining authors declare that the research was conducted in the absence of any commercial or financial relationships that could be construed as a potential conflict of interest.

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
