# Peer review of "Effect of Transplanting Time and Nitrogen–Potassium Ratio on Yield, Growth, and Quality of Cauliflower Landrace Gigante di Napoli in Southern Italy"

_horticulturae, doi:10.3390/horticulturae10050518_

Round 1
Reviewer 1 Report
Comments and Suggestions for Authors
It is interesting and useful that the authors have investigated effect of transplanting time and nitrogen-potassium ratio on yield, growth and quality of cauliflower. In total, the MS was written well. Hence, it is recommended to be accepted after minor revisions.
1. Add amounts of mean temperature and total rainfall in the text, but delete Fig.1,
2. Add sub-section in the section of results and discussions.
3. Change “ppm” to “mg kg-1 in both Table 2 and the text, due to ppm being no long an international unit.
Comments on the Quality of English LanguageMinor editing of English language is required.
Author Response
Dear Reviewer, thanks a lot for contributing to improve the quality of our manuscript. We have addressed your comments across the text, highlighting the modifications/amendments with the red colour, and reported below the related answers.

Reviewer 2 Report
Comments and Suggestions for Authors
This study investigated the effects of transplanting time and N:K ratio on the growth, yield, and quality of cauliflower plants in southern Italy. Certain results in the Results and Discussion section are inconsistent with the data or statistics presented in the Tables; please refer to the specific comments below. Additionally, there appears to be redundant information in the discussion; consider consolidating and summarizing it.
Line 150: Do you mean “…was homogenized for 30 min”?
Line 206-207: The statement “The earlier transplanted crops exhibited a longer growing period, compared to the later one (81.4 vs. 73.1 days)…” is not consistent with the data presented in Table 1. Please verify.
Line 305-306: According to Table 2, there was significant difference in K levels between the two transplanting dates.
Line 335-336: Reiterating the abbreviations is unnecessary.
Line 407: Italicize “Helianthus annuus”.
Line 410: Italicize “Brassica napus”.
Author Response

(The authors gave the same response as above.)

Reviewer 3 Report
Comments and Suggestions for Authors
The topic addressed by the authors of the manuscript is interesting for the wider scientific community. The topic is challenging and demanding. Unfortunately, the authors did not respond to this challenge.
Significant improvements need to be made in the manuscript. The following shortcomings were observed, which need to be corrected, in order for this manuscript to meet the high standards of publishing papers in the journal.
Title: satisfies.
Abstract: satisfies.
Keywords: inappropriate choice of keywords: Biodiversity, nitrogen use efficiency (NUE).
Introduction: satisfies.
Materials and Methods:
Line 102-104. Оn which type of soil the research was carried out (according to the FAO soil classification)?
Line 119-120. Why is the sentence separated?
Line 127-132. Sentence too long (in 6 lines!). Give an explanation: why exactly these parameters are measured? And indicate the appropriate literature source for the procedure (methods) of these measurements.
Line 193-197. The formula for calculating the NUE value is not appropriate. In agrochemistry, the NUE value is obtained from the ratio: assimilated amount nitrogen by the crop / applied amount of nitrogen. The same applies to the KUE value.
Line 199-201. Give an explanation: Why were the correlations between the individual studied parameters not analyzed?
Results and Discussion:
According to the Material and Methods chapter, the results and discussion should be presented in 7 subsections (it is more practical and clear for reading and understanding).
Line 203-226. Interpretation of the obtained results very modest and not adequately supported by cited references.
Line 401-417. The use of the term "nitrogen use efficiency" (as well as the term: "potassium use efficiency") is highly debatable. It would be more appropriate to use the term (for example): "biological yield index of the amount of applied nitrogen (potassium)".
Conclusions: satisfies.
References: Of the total number of cited references (54), 10% of them were published in the last 2 years. However, even 1/3 of the cited references are older than 20 years. Citations of some of these references are considered common knowledge and may be omitted.
Author Response

(The authors gave the same response as above.)
